**NHESS Brief Communication**
**Mountain roads in Nepal at a new crossroads**
Dr. Karen Sudmeier-Rieux, Institute of Earth Sciences, Faculty of Geoscience and Environment,
University of Lausanne, Switzerland
Prof. Brian G. McAdoo, Yale-NUS College, Singapore
Mr. Sanjaya Devkota, Institute of Engineering, Tribhuvan University, Nepal
Mr. Purna Lal Chandra Rajbhandari, Independent, Kathmandu, Nepal
Mr. John Howell, Consultant, Living Resources Ltd., Devon, U.K.
Mr. Shuva Sharma, Scott Wilson Nepal Ltd., Kathmandu, Nepal

**Introduction – roads as vehicles of development?**
For the past two decades, development of the road network in Nepal has topped community and
government priorities, a trend likely to continue as the country transitions to a decentralized Federal
government (Rankin, 2017). In parallel, China's new Belt and Road Initiative (BRI) offers the promise of
investments in key infrastructure: expanding trunk roads, hydro-electricity, trade and development
(The Wire, 2017; The Economist, 2017).  Yet as Nepal devolves significant power to local and Provincial
administrations, it is uncertain whether the newly formed local administrations will rise to the
challenge of establishing safeguards to ensure that promised benefits outweigh potential losses. We
suggest that the issue of poor roads in Nepal is a political, not a technical issue and one where better
service and less environmental damage could both be significantly addressed through improved
governance. This commentary points to the need for improved road governance based on research,
consultations and observations of road construction and associated landslides in Nepal.  It also
highlights the need for more scientific studies on the topic as most relevant publications emanated
from the grey literature, government publications or media articles.
Roads are globally accepted livelihoods links for communities in rural areas.  By reducing travel time
on foot, opportunities are opened for quicker transportation of goods, better access to employment,
education and health (Bryceson et al., 2008; Hettige, 2006; ).  Roads generally create direct and indirect
benefits to rural populations, directly through employment in constructing and maintaining roads, and
in providing rural transportation services. Indirectly they provide opportunities for marketing goods
and services, flexibility for employment and roadside businesses, and for transporting agricultural
products to markets (Bryceson et al., 2008; Iimi et al., 2016).  They can provide a safety net of sorts in
generating alternative livelihood opportunities, especially in circumstances where conditions for
agriculture are difficult. In general, connectivity is thus positively correlated with lower poverty rates
(Hettige, 2006; Iimi et al., 2016). Additionally, there are many non-monetary benefits of roads,
especially greater access for the poor to health and other public services, such as education, which can
significantly reduce vulnerability and even gender inequality (Starkey et al., 2013). In Nepal, roads are
also linked to the current boom in migration, facilitating easier mobility to both near and distant
migration destinations (Jaquet et al., 2015; Upreti and Shrestha, 2015). Finally, a robust road
infrastructure can provide vital corridors for evacuation and rescue in the aftermath of disaster.
However, benefits of roads need to be weighed alongside evidence that roads may benefit non-poor
households more, perhaps making development less even (Hettige, 2006). Furthermore, other
impacts, such as increased environmental hazards, pollution, crime and unwanted cultural influences
are often overlooked (Blaikie et al., 1976; Hettige, 2006; Murton, 2016; Jaboyedoff et al., 2016). This
manuscript builds on research and publications questioning the aspirations of the Government of
Nepal as early as the 1970s and 1980s. The Overseas Development Group at the University of East

Anglia pioneered studies to understand short-, medium- and long-term effects of road construction on spatial and socio-economic inequality (Blaikie et al., 1976; Rankin et al., 2017). Blaikie, Cameron and Seddon (1980) revealed the inequalities created by road construction, with loss of livelihoods for those without possibilities to invest, and enhanced opportunities for those who could (Rankin et al., 2017).

This work was conducted during the same period as the Laban (1979) benchmark inventory of landslides in Nepal to document the number of landslides and their origin as either natural or human-induced. Although roads represented a small proportion of total land area at the time, Laban warned that as the road network continued to expand, the number of landslides will, "increase drastically in the near future, especially if more careful construction methods are not undertaken" (Laban, 1979: iv). Both research projects were widely influential and according to Rankin et al (2017), the Blaikie et al (1976) study may have redirected domestic budgets and foreign aid toward other rural development investments. However, this reprieve was soon to end with a greater focus on connectivity in the 10th 5-year plan (2002-2007) and the boom in foreign investments in road construction projects after 2008 and the end of the Maoist insurgency (Pokharel and Acharya, 2015). The 11th plan (2007-2010) established the ambitious goal of constructing a road network throughout the country whereby residents in the Hills should have a road available within four hours walking distance and Terai residents within two hours (Pokharel and Acharya, 2015.

**Nepal's mountain roads – vehicles of disaster?**

Roads in Nepal are generally classified as national roads, (i.e. Strategic Road Network, SRN) under the jurisdiction of the Department of Roads (DOR), or local roads (i.e. Local Road Network, LRN). The LRN is comprised of District Road Core Network (DRCN) and Village Roads (VR) under the jurisdiction of the Department of Local Infrastructure Development and Agricultural Roads (DOLIDAR) (Figure 1). Road building started to gain momentum in Nepal with the advent of multi-party democracy in the early 1990s, intensified further after the Maoist insurgency ended in 2006 and continues to be one of the country's main priorities (Upreti and Shrestha, 2016; DOLIDAR, 2016a).

73

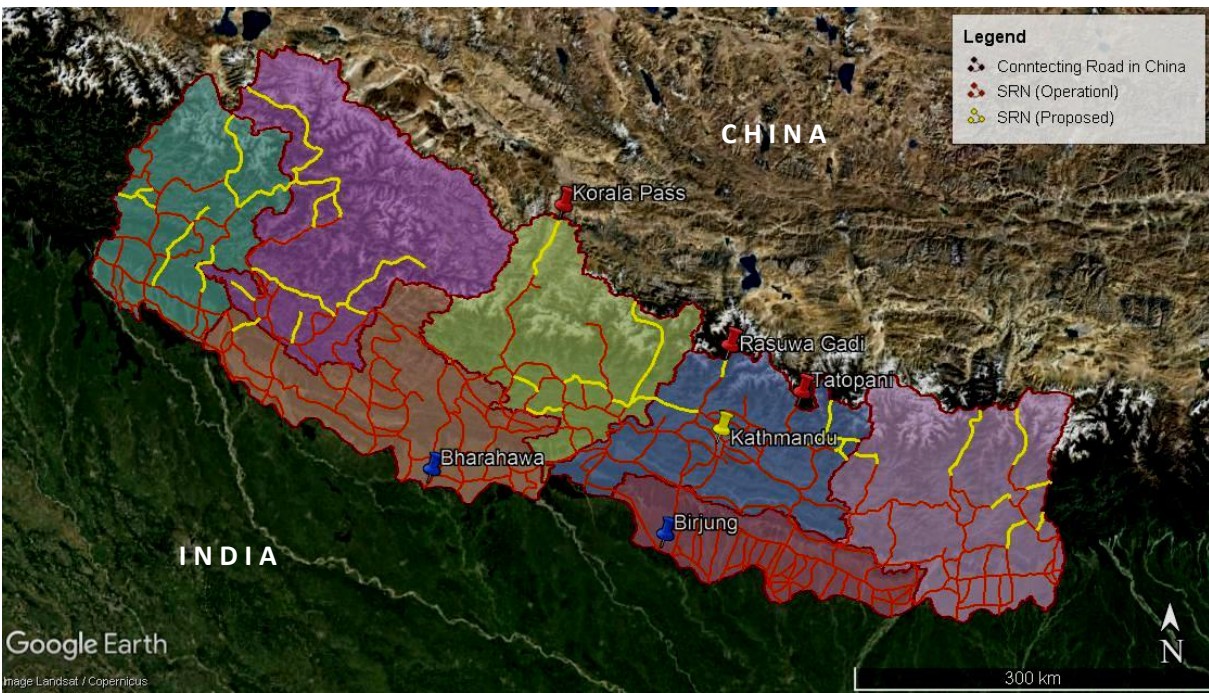

**Figure 1. Nepal Provincial boundaries and national (SRN) road network highlighting existing and proposed roads, according to DOLIDAR (2016) and main current border crossings with China and India. As Provincial administrations are in the process of revising Master Plans to represent new administrative boundaries, there is opportunity to put in place governance mechanisms for improved road construction and maintenance. (Source: Modified from DOLIDAR, 2016, based on Google Earth imagery).**

Twenty years ago, Nepal's road network was one of the lowest in the world with a road density for both SRN and LRN estimated at 13.7 kilometers (km) per 100 $km^2$ in 1998 (DOR, 2002; DOR, 2017). By 2016, it had increased to 49.6 km per 100$km^2$ and continues to increase at a very rapid pace (DOLIDAR, 2016a). The SRN expanded rapidly from 4,740 km (blacktop, gravel and earthen) in 1998 to 15,404 km in 2016 (DOLIDAR, 2016). The LRN experienced a 1200 percent increase during this period, from 4,780 km in 1998 to 57,632 km in 2016 and are the most common roads in rural areas (DOLIDAR, 2016a).

In 2007, the country spent 5.2 percent of its national budget on roads, but by 2011/12 this figure had increased to 6.7 percent or an estimated 491.2 million USD (WB-GON, 2014). The estimated investment in the LRN was about 245.6 million USD (2011/12), of which 54 percent of the rural road budget originated from donors and 20 percent were soft loans to communities. Community contributions amounted to an estimated 12 percent of the total budget through their own savings and remittances, and earnings from community forestry (WB-GON, 2014; DOLIDAR, 2016b). This demonstrates the significance and priority given to roads and connectivity as a vector for economic development and population mobility.

Despite the budget and priority allocated to the road network, Nepal's mountain roads are in a treacherous state, subject to frequent rockfall, landslides and accidents (Singh, 2018; DoR, 2013a) (Figure 2). According to DoR (2013a), one of the main causes of road accidents is road design, including very steep gradients, lack of safety features and poor road conditions. Local road construction or so-called 'dozer roads' are most often initiated and constructed by bulldozer owners in collaboration with politicians at the request of communities, without basic grading or drainage (ITAD, 2017; Singh, 2018).

The dozer roads are usually constructed or upgraded during the dry season. During the monsoon, road
segments are frequently washed out because a majority of these roads lack proper engineering (WB-
GON, 2013).  Road failures are cleared up at high cost after the monsoon and the failure-and-clearance
process is repeated for years until there is no loose soil to block roads (Leibundgut et al., 2016).
Environmental impacts include destroyed irrigation schemes, springs and contaminated water supplies
(Singh, 2018). Initial Environmental Examinations (IEE) to reduce environmental impacts are usually
required for local road construction but are rarely enforced (ITAD, 2017).

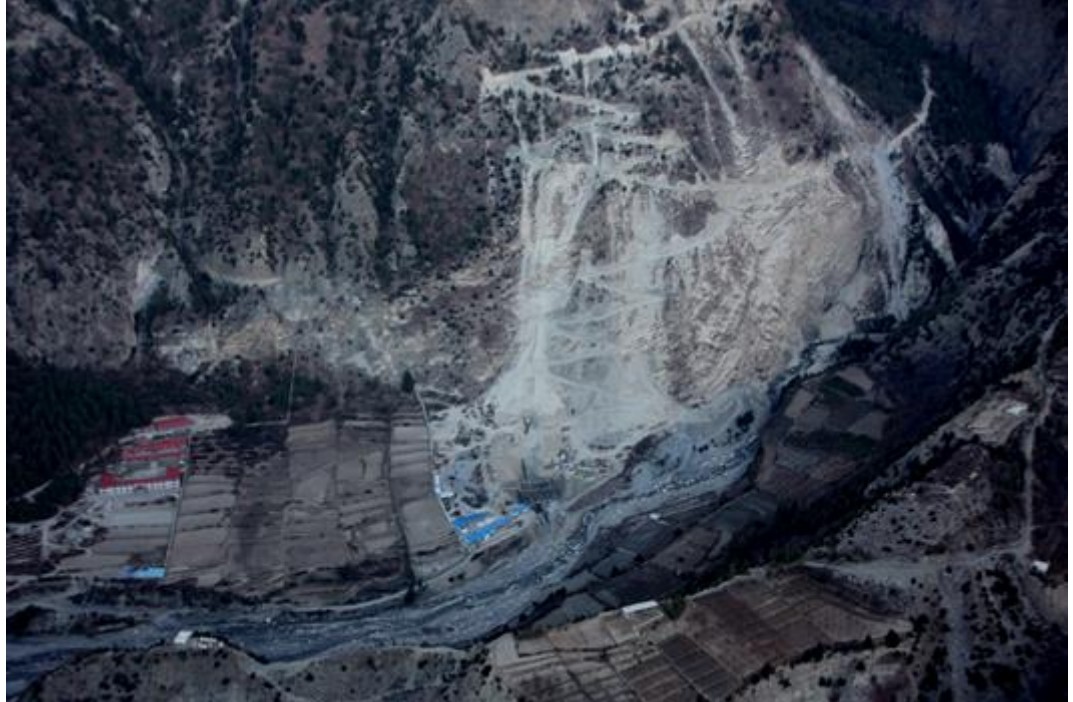

**Figure 2. Local road, Lower Mustang District, Province 4, Nepal. Credit: Rajbhandari, 2016.**

Such rapid and ineffective road construction throughout the country, but particularly in the middle hill
and mountain areas, is placing increasing pressure on fragile ecosystems, wasting government
resources and increasing risk to road passengers and roadside dwellers (DoR, 2013a; Singh, 2018).
Studies have demonstrated that roads are one of the greatest anthropogenic drivers of environmental
degradation, erosion and landslides in Nepal (Leibundgut et al. 2016; Froude and Petley, 2018; McAdoo
et al, accepted ; Petley et al., 2007; Vuilliez et al. 2018).  This situation is worsening due to the
intensifying rainfall during the monsoon, largely attributed to climate change (Bharti et al., 2016;
Devkota et al. 2018; Froude and Petley, 2018; Petley et al., 2007), which has led to a greater occurrence
of landslides, especially in the middle hills (McAdoo, accepted).  The possibility of an earthquake of
even greater magnitude than the 2015 Gorkha earthquake (M 7.8) raises concerns about poorly
designed roads increasing the likelihood of catastrophic landslides (Singh, 2018).

**Nepal at a governance crossroads**
Nepal has a range of acts, regulations, guidelines and directives that require proper road engineering
practices, various levels of environmental assessments and approval.  However, while funded by
government budgets, a majority of local roads do not follow established government practices (ITAD,
2017). Hence, although the legal framework for ensuring proper governance of infrastructure
development is well developed with public bodies to monitor and enforce governance, the lack of
political will and consensus among political leaders has undermined the impact of these bodies (WB-
GoN, 2013).
As the country shifts decision-making to the Provinces, it is unclear how management of roads will be
affected among the main actors such as DOR, DoLIDAR, and rural and urban Municipalities.
Institutional roles are shifting under on-going reforms, with executive authority over local
infrastructure development being transferred from district level authorities (District Technical Officer)
to Provincial Public Works Departments, which are supposed to coordinate with central level ministries
and departments (ITAD, 2017). At the time of printing, it is not yet clear which administrative body
will have oversight of road policies and alignment of policies between Provinces. The risk is that the
few gains that had been achieved over the past decade, including a greater emphasis on regular
maintenance of roads, become completely diluted (ITAD, 2017).
Another development which may affect the type and pace of road construction in Nepal is China's Belt
and Road Initiative (BRI). In May 2017, Nepal became a signatory to the BRI with the promise of
expanding several trunk roads in order to foster new trade and economic benefits (The Economist,
2017). This new "Silk Road" will develop a trade and infrastructure network from China towards the
west and south including countries in Central and South Asia and Eastern Europe.
The BRI has for now elicited more questions than answers, including: which roads will be expanded,
will it link rural mountain communities to greater economic development opportunities, better health
care and education options, and increased social networks; or will the BRI trunk roads spawn more of
the poorly engineered local roads with their demonstrated low cost effectiveness and high
environmental impacts? Without adequate controls and support, rural villages can be expected to tie
into these trunk roads by expanding the network of poorly-constructed local roads, with ensuing
environmental, economic and human risks associated with roadside erosion and slope failures that
damage both the roads and the neighboring productive land.
Despite this bleak picture, Nepal has the governance systems in place to resolve the problem if it
chooses to do so. Numerous technical manuals and departmental guidelines provide the basis for good
alignment determination, careful engineering, the stabilization of incipient landslides in slopes and the
prevention of erosion through the use of bio-engineering (Deoja, 1994; DOR, 2013b). Nepal has been
a world leader in the past and government agencies such as DOR and DOLIDAR all have cadres of highly
trained engineers and bio-engineers who could fulfill the required technical functions satisfactorily if
directed properly (ITAD, 2017; WB-GON, 2013).
However, these abilities are currently ignored in the interest of political expediency and a misplaced
public perception that quickly opened roads are a panacea for socio-economic development.
Institutions were established to regulate road construction. The Environmental Protection Council was
formerly established under the Chairmanship of the Prime Minister to monitor environmental impacts
and to regulate the environmental and social impact assessment legal instruments (GON, 1997), but
became ineffective facades. The Department of Roads' Geo-environmental and Social Unit is also not
serving its function. Finally, political influence has overrun any efforts to instill checks and balances
(ITAD, 2017), notably by the Commission for Investigation of Abuse and Authority, which was created
to highlight cases of poor governance.
Yet with the revision of ministerial portfolios in 2018, the re-organized Ministry of Forests and
Environment has an opportunity to ensure that statutory environmental safeguards are met by those
government units that will be responsible for administering road development. Newly formed
Provincial administrations are now tasked with revising their Master Plans and have the opportunity
to develop action plans to strengthen governance bodies, increase transparency and enforce
regulations.

**Conclusions**

On the surface, roads are vital livelihood links for rural populations for improved access to markets,
health care, education, employment and migration. Mobility is increased, rural populations can
develop greater resilience to harsh environmental conditions, and there are possibilities of new
economic opportunities, ultimately reducing economic vulnerability.  However, mountain roads,
especially when poorly constructed, present particular challenges of sustainability, risk and
governance (Sidle and Ziegler, 2012).  Hence, the full benefits of such roads in mountainous areas
should be questioned.
Finally, the issue of poorly designed and risk-filled roads in Nepal, is a political, rather than technical
issue.  As Nepal moves towards greater decentralization of power, there is considerable opportunity
for its local and national administrations to turn the tide toward safer and more sustainable road
development. The two new major drivers of road development in Nepal – decentralization of power
and the BRI – could be harnessed to change road construction from the current trajectory of
environmental disaster to vectors for development. The high environmental and maintenance costs of
haphazard 'dozer roads' could be significantly reduced if government policies were enforced to achieve
well-established road engineering designs, including basic standards of road grading, alignment,
drainage and bio-engineering.  Nepal is at a new crossroads with fresh opportunities to rein in the
"dozer road" constructors, but this will require concerted effort and considerably more political will
power than has been demonstrated over the last decade.

**Acknowledgements**

The authors would like to thank Prof. Michel Jaboyedoff, Faculty of Geosciences and Environment,
Institute of Earth Sciences at the University of Lausanne, the Ecosystems Protecting Infrastructure and
Communities (EPIC) project, managed by the International Union for Conservation of Nature, funded
by the German Federal Ministry for the Environment, Nature Conservation, Building and Nuclear Safety
(BMUB) 2012-2017, for contributing toward this research along with funding from Yale-NUS College's
Centre for International and Professional Experience.   We also thank anonymous reviewers and the
guest editor for suggestions which significantly improved this manuscript.

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

Index: Main Determinants and Correlations to Poverty. Transport and ICT Global Practice, The World
Bank Group, November 2016, Policy Research Working Paper 7876. Washington D.C.: the World Bank,
18 pp. 2016. https://openknowledge.worldbank.org/handle/10986/25676 2016.
Jaquet, S., Schwilch, G., Hartung-Hofmann, F., Adhikari, A., Sudmeier-Rieux, K., Shrestha, G., et al.: Does
outmigration lead to land degradation? Labour shortage and land management in a western Nepal
watershed. Applied Geography, 62, 2015, 157–170. 2015.
Jaboyedoff, M . Michoud, C. Derron, M-H. Voumard, J. Leibundgut, G. Sudmeier-Rieux, K. Nadim, F. and E.
Leroi: Human-induced landslides: toward the analysis of anthropogenic changes of the slope
environment. Landslides and Engineered Slopes. Experience, Theory and Practice Proceedings of the
12th International Symposium on Landslides (Napoli, Italy, 12-19 June 2016) Edited by Stefano Aversa,
Leonardo Cascini, Luciano Picarelli, and Claudio Scavia. 2016.
Laban, P.: Landslide occurrence in Nepal. Kathmandu: Integrated watershed management torrent control and
Land use development project, Ministry of Forest, Department of Soil and Water Conservation and the
Food and Agriculture Organisation. 1979.
McAdoo, B. Quak, M., Sudmeier-Rieux, K., Gnyawali, K., Adhikari, B., Devkota, S., Rajbhandari, P.: (accepted for
publication). Roads and landslides in Nepal: How development affects risk. Submitted to NHESS, 31
December 2017.

Murton, G.: A Himalayan Border Trilogy: The Political Economies of Transport Infrastructure and Disaster
Relief between China and Nepal. Cross-Currents: East Asian History and Culture Review E-Journal
No.18, March, 2016.
Petley, D., Hearn, G.J., Hart, A., Rosser, N., Dunning, S., Oven, K., Mitchell, W.: Trends in landslide occurrence in
Nepal. Nat Hazards, 43: p. 23–44. 2007.
Pokharel, R. and S.R. Acharya.: Sustainable Transport Development in Nepal: Challenges,
Opportunities and Strategies. Journal of the Eastern Asia Society for Transportation Studies, Vol.11,
2015.
Rankin, K.N., Sigdel, T.S, Rai, L. Kunwar, S. and P. Hamal.: Political Economies and Political Rationalities Of
Road Building In Nepal, Studies in Nepali History and Society 22(1): 43–84 June 2017
Sidle R.C. and Ziegler A.D.: The dilemma of mountain roads. Nature Geoscience, 5, 437-438. 2012.
Sidle, R. C., Ghestem, M., and A. Stokes.: Epic landslide erosion from mountain roads in Yunnan, China –
Challenges for sustainable development.  Nat Hazards Earth Syst Sci 14, 3093-3104. 2012.
Singh, B.P.S.: From nowhere to nowhere. Haphazard road construction is ravaging the Nepali countryside.
Nepali Times, July 6, 2018. https://www.nepalitimes.com/banner/from-nowhere-to-nowhere/ 2018.
Schwilch, G., Adhikari, A., Jaboyedoff, M., Jaquet, S., Kaenzig, R., Liniger, H. P., I. Penna, K. Sudmeier-Rieux and
B.R. Upreti.:  Impacts of outmigration on land management in a Nepali mountain area. In K. Sudmeier-
Rieux, M. Jaboyedoff, M. Fernandez, I. Penna, & J. C. Gaillard (Eds). Identifying emerging issues in
disaster risk reduction, migration, climate change and sustainable development - shaping debates and
policies. Springer Publisher. 177–196. 2016.
Starkey, P., Tumbahangfe, A., Sharma, S.: Building roads and improving livelihoods in Nepal, External
review of District Roads Support Project: Final Report. Swiss Agency for Development and Cooperation
(SDC), District Roads Support Programme (DRSP) Kathmandu: SDC, 82 pp.
https://doc.rero.ch/record/255566/files/46-External_Review_District_Roads.pdf 2013.
The Economist.: The two sides of the mountain. December 23, 2017.
https://www.economist.com/news/asia/21732851-maldives-nepal-and-sri-lanka-are-no-longer-meek-
they-used-be-india-faces-growing  2017.
The Wire.: One Belt, One Road Fuels Nepal's Dreams. July 11, 2017.
https://thewire.in/156554/nepal-china-obor-transport-infrastructure/ 2017.
Upreti, B. R., and Shrestha, G.: Linking migration, mobility, and development for strengthening adaptation to
climate and disaster Risks: Reflections from Nepal. In K. Sudmeier-Rieux, M. Jaboyedoff, M. Fernandez,
I. Penna, & J. C. Gaillard (Eds). (2016) Identifying emerging issues in disaster risk reduction, migration,
climate change and sustainable development - shaping debates and policies. Springer Publisher, 146-
160. 2016.
Vulliez, C, Tonini, M, Sudmeier-Rieux, K, Devkota, S. Derron, M-H, and M. Jaboyedoff, M.: Land use changes in
the Phewa Lake Watershed of Western Nepal. A comparative study 1979-2016. Journal of Applied
Geography 94. 30-40. 2018.
WB-GON [World Bank, Government of Nepal]. Nepal Road Sector Assessment Study. Kathmandu; WB-
GoN, 54pp. http://rapnepal.com/sites/default/files/report-
publication/road_sector_assessment_study_main_report_final_30may2013.pdf 2013.