# Peer review of "NHESS Brief Communication"

_Natural Hazards and Earth System Sciences, 2017_

## Referee Comment (RC1) · Anonymous Referee #1 · 14 Feb 2018

General Comments:

This brief communication discusses rural road construction in Nepal and the effect on landslide occurrence, with a wider overview pertaining to the geopolitics of this complex region, particularly the role of China. Such a topic is important and deserves discussion however, effectively addressing the many varied and complex issues in such a short manuscript is difficult, and to my view, this manuscript does not do a sufficient job of addressing the issues at hand. This manuscript falls down at the very outset with the list of questions presented at the end of the introduction. The importance of the questions posed is in no doubt, however to address each sufficiently requires far

more than a brief communication, and certainly not one with a sole focus on road-induced landslides. The authors appear stuck between writing an historic summary of road building policy and its effects and providing an over-arching discussion about the difficult geopolitics of the region. Certainly, there is no novel scientific contribution provided; the links between rural road construction and landslides has been discussed and demonstrated by many authors previously, some of which the authors reference. Thus, the manuscript could be significantly improved by refocussing the paper and being clear from the outset what is the novel contribution being made.

A further issue is the oversimplification throughout the paper. This comes about partly through the desire to reduce length and maintain a 'brief communication' - however the topic is not one that gives itself to a short discussion. For instance, do the authors really feel that the appropriate way to address questions such as "who will win and lose" between two superpowers (line 36-37), and "will Nepal rise to the challenge of establishing safeguards to ensure promised benefits outweigh losses as it transitions to the newly established federal system?" (line 40-42) is simply by discussing well established links between road construction and landslides and not the myriad of other issues involved? Is a better question, and one significantly easier to answer and arguably more directly relevant - will the increased landsliding resulting from these roads outweigh the benefits they bring? I would urge the authors to consider what is they really wish to address and discuss in this manuscript and focus on that. Trying to deal with such large-scale geopolitical questions is to be commended, but to do so sufficiently cannot be from such a narrow and specific view point.

Specific comments:

Abstract: This reads more like an introductory statement than an abstract, and points to the fact the manuscript has little new scientific results to present. In fact, it is exactly duplicated from the first paragraph of the Introduction, for which it is better suited.

Line 30: recent landslide victory - confusing to use landslide in this context when the

rest of the paper focusses on the physical phenomena

Line 36-42: None of these questions are actually addressed in the paper - probably because its not really possible to do so from such a narrow view point

Line 42: based on research - your own or others?

Line 50: road construction has been closely linked to nation-building - but this is just one of many factors, right?

Line 56-58 - Not clear what this has to do with road construction?

Line 63-65 - One of the many simplifications in this manuscript - the signing of the new constitution and the subsequent blockade spun out of many varied and complex internal and external factors. You mention the complex factors of Indian discontent and ethnic sympathies, but ignore the internal politics in India that were also leveraged by the blockade.

Line 67-68 - Again, China's relationship with Nepal is complex and to suggest it is good due to the response to the earthquake is to over simplify the issue. China repeatedly clashed with Nepal and India during the response over air space and Indian military close to the border. At one stage China threatened to pull out unless India and Nepal played by its rules. Nepal siding with China was another causative reason for the subsequent Indian blockade. These few lines here just emphasise the over reach of this manuscript - these issues themselves require much more discussion and do not just come down to road building and landslides.

Line 79-81 - but this infrastructure is also badly needed. Connecting isolated communities enables socio-economic development that can greatly increase development for isolated rural populations.

Line 83 - proper engineering standards - debatable, many of these roads are constructed to the standards of the host country, which are often grossly insufficient. Sections of the Arniko are a perfect example.

Line 83-87 - this is certainly an issue, but also ignores a further issue that was seen in the 2015 earthquake. These main roads become economic belts which drive migration to form new road-side settlements (e.g. Chaku on the Arniko). These roads are typically built in valley bottoms to make construction easier meaning that populations migrate from what were previously reasonably landslide-safe locations down into valley bottoms where their exposure to landslides dramatically increases. Many of the fatalities in 2015 in Sindhupalchowk occurred amongst new road-side communities. So while this road construction undoubtedly increases hazard it also increases population exposure.

Line 91-93 - do you have a reference to support this?

Line 95 - in the middle and lower hills - but above you mention that road construction is increasing mainly in the high and mountain areas. This would suggest the increased landsliding in the Siwaliks is related to stronger monsoons and not road construction?

Line 126-127 - precisely, what will the consequences be for Nepal? You do not directly answer this question anywhere.

Line 151 - this isn't the sole reason referred to by Petley et al 2007 - they also highlight that land use changes (i.e. urbanisation) and the now-ended Civil War as being other key contributors.

Line 158 - "associated with a road" - what does this mean exactly? Did the road cause the landslide, or is it just that a landslide occurred near a road? Defining what you mean by 'associated with' is important.

Line 165 - "ridge shaking effect" - I'm not sure there is enough evidence to conclusive determine why coseismic landslides are different to rainfall landslides just yet. Many papers point to numerous different factors involved in coseismic landsliding, with the ridge effect being just one. For instance see:

Yamagishi, H., & Iwahashi, J. (2007). Comparison between the two triggered landslides

in Mid-Niigata, Japan by July 13 heavy rainfall and October 23 intensive earthquakes in 2004. Landslides, 4(4), 389-397.

Chang, K. T., Chiang, S. H., & Hsu, M. L. (2007). Modeling typhoon-and earthquake-induced landslides in a mountainous watershed using logistic regression. Geomorphology, 89(3-4), 335-347.

Van Westen, C. J., Van Asch, T. W., & Soeters, R. (2006). Landslide hazard and risk zonation—why is it still so difficult?. Bulletin of Engineering geology and the Environment, 65(2), 167-184.

Line 176 - "after 12 years" - that seems an awfully long time for a rural population struggling for access and the associated economic benefits right now.

Line 190 & 194 - repitition

Line 192 - typo, 20th

Figure 1 - I'm not sure what the isoseismals in the inset are supposed to add, especially since the earthquake is only mentioned once in the text. In any case, they are difficult to distinguish with the present colour scheme. Likewise for rainfall, I'm not sure what this adds given there is no specific discussion of varying rainfall amounts over Nepal. Would it not be better to show the population distribution to make clear the number and importantly location of people surrounding these roads?
* * *

---

## Referee Comment (RC2) · Anonymous Referee #2 · 8 Mar 2018

The topic of this study is timely and important but the article as it currently stands does not offer significant new knowledge, conclusions, or actionable insights. While a clear connection between roads and landslides clearly exists, the scope of the article is far too wide and ambitious, such that very little traction is made with respect to any of the multiple topics addressed. The methodology of data collection and analysis also remains unclear, and appears to be a combination of popular media accounts in the BRI contexts with historical studies of land degradation (UEA study) and a paucity of primary fieldwork data. All this being said, this reviewer finds real and important potential in the study and encourages the authors to undertake a significant rewrite that focuses the paper in more direct ways (for example, this could and maybe should be more

about roads and landslides than BRI and the politics of post-earthquake development). A single article does not (and cannot) address everything all at once!

More specifically, items of consideration include:

Lines 36-43: Second paragraph - important questions but overly simplified and too much to answer in this brief commentary

Lines 60-68: Too simplistic - road and infrastructure development and state making; legacies of Indian hegemony; earthquake; constitution and fuel blockade; Chinese humanitarianism; and BRI is TOO MUCH for one paragraph (and probably one paper)

Lines 80-81: This is an understatement and underdeveloped consideration with respect to the BRI - "Infrastructure in such mountainous terrain warrants special attention to ensure that construction does not cause environmental harm or aggravate disaster risks such as landslides and flooding." High number of references are from online and popular media sources. Where is the data?   Page 4 - Landslides and roads - Peter Laban etc. history of land degradation and landslides is important topic. But introduced suddenly and paid insufficient attention

Lines 147-152 - Statistics of road-based fatalities are tragic. But these numbers do not account for the exponentially higher number of travelers on roads today. Nor do they account for many landslides that are accounted for differently in non-road hillsides

Lines 162-167 - so is conclusion that there is no reliable connection between roads and increased landslides; more of an issue of monsoon, etc.? Please clarify this point.

Line 198 - Strongly disagree with this statement. Nepal may have good technological skills and capable people, but the natural and objective challenges of roadbuilding in the Himalaya coupled with protracted corruption and compromised leadership compound the delays, hazards, and dangers of road development in Nepal. That is, in fact it very much IS a political issue.

---

## Editor Comment (EC1) · F E Taylor (Editor) · 4 May 2018

Thank you to anonymous reviewer 1 and 2 for their constructive and helpful comments in their review of NHESS-2017-462. Both reviewers have commented on the timeliness and relevance of the paper, but that it tries to answer too many questions in a short space. Indeed, there are perhaps several research papers condensed into one brief communication here. To move forward, the authors must focus the paper. I have been in discussion with the lead author Karen with regard to this, as there are several directions in which the paper could be taken.

We have agreed that the paper should remain as a brief communication type (b) as

defined by NHESS as: "report/discuss significant matters of policy and perspective related to the science of the journal, including "personal" commentary". The focus of the paper should be on the contemporary policy issues surrounding the belt and roads initiative and its implications for hazards. This means reducing some of the historical descriptions and some of the review of previous literature discussing evidence of the interaction between roads and landsliding in Nepal. This shifts the paper from being primary research to a short, opinion piece that informs the NHESS readership about the policy developments and stimulates new research around this topic.

Both reviewers have made very useful comments with regard to ensuring the commentary is not too reductionist or one-sided. Due to the more focused nature of the revised paper, there should be more room to ensure these are implemented. A small number of these comments may no longer be relevant with the new focus of the paper.

Thank you again to Reviewer 1 and 2 for their input. I look forward to the author responses to reviewers.

---

## Author Comment (AC1) · 20 Jun 2018

The authors would like to thank the two anonymous reviewers for their careful reading of our commentary, "Brief Communication: Vehicles for development or disaster? The new Silk Route, landslides and geopolitics in Nepal". In light of the editor's suggestion to refocus the paper, we will not respond to each reviewer point but rather provide an overall response. The main point made by both reviewers and the editor is that the commentary tried to answer too many questions in a short space, with several research papers condensed into one brief communication. This point is well taken and the authors have accepted to undertake a major rewrite to a shorter opinion piece. The

focus will thus be more policy-oriented with the objective of highlighting key environmental and governance issues with road construction in Nepal in light of several new drivers, in particular, the new decentralized Federal governance structure and the Belt and Road Initiative (BRI).

The authors will, however, respond to a few specific comments: Reviewer 1 "the links between rural road construction and landslides has been discussed and demonstrated by many authors previously, some of which the authors reference" There are a handful of studies which have demonstrated this relationship based on research from around the world, but few or none to our knowledge, in Nepal since the 1979 Laban study. The fact that 74% of all landslides were natural was true in 1979 but no longer today. Yet this figure is still used in Nepal and has even become a finger-pointing political issue. Thus, the importance for more empirical research on anthropogenic causes of landslides, including roads, in Nepal.

Reviewer 2 – With reference to the BRI; "High number of references are from online and popular media sources. Where is the data?" This comment is interesting and the response is that as the issues surrounding the BRI are so recent, there are hardly any peer-reviewed publications on this topic, at least not according to our research. Thus the novelty and importance to research and publish on this topic, which is what we are intending with the revised commentary.

Thank you Karen Sudmeier-Rieux
* * *

---

## Author Comment (AC2) · 14 Aug 2018

The authors would like to thank the anonymous reviewer for their careful reading of our commentary, "Brief Communication: Vehicles for development or disaster? The new Silk Route, landslides and geopolitics in Nepal". In light of the editor's suggestion to refocus the paper, we will not respond to each reviewer point but rather provide an overall response.

The main point made by both reviewers and the editor is that the commentary tried to answer too many questions in a short space, with several research papers condensed into one brief communication. This point is well taken and the authors have accepted

to undertake a major rewrite. The new article is an opinion piece with the objective of highlighting key environmental and governance policy issues with regards to road construction in Nepal in light of several new drivers, in particular, the new decentralized Federal governance structure and the Belt and Road Initiative (BRI).

The authors would however like to respond specifically with regards to: Reviewer 1 "Certainly, there is no novel scientific contribution provided; the links between rural road construction and landslides has been discussed and demonstrated by many authors previously, some of which the authors reference. Thus, the manuscript could be significantly improved by refocussing the paper and being clear from the outset what is the novel contribution being made."

There are a handful of studies which have demonstrated this relationship but few, if none, in Nepal since the 1979 Laban study. The fact that 74% of all landslides are natural was most likely true in 1979 but no longer today, yet this figure is still used in Nepal and has even become a political issue. The country is also at a new crossroads with a major government reshuffle and new opportunities provided by the BRI

Thus the novelty and importance to research and publish on this topic, which is what we are intending with the revised commentary. We hope the rewritten brief communication meets expectations. Thank you once again to NHESS for considering our revised manuscript for publication.

---

## Author Comment (AC3) · 14 Aug 2018

The authors would like to thank the anonymous reviewer for their careful reading of our commentary, "Brief Communication: Vehicles for development or disaster? The new Silk Route, landslides and geopolitics in Nepal". In light of the editor's suggestion to refocus the paper, we will not respond to each reviewer point but rather provide an overall response. The main point made by both reviewers and the editor is that the commentary tried to answer too many questions in a short space, with several research papers condensed into one brief communication. This point is well taken and the authors have accepted to undertake a major rewrite. The new article is an opinion

piece with the objective of highlighting key environmental and governance policy issues with regards to road construction in Nepal in light of several new drivers, in particular, the new decentralized Federal governance structure and the Belt and Road Initiative (BRI).

The authors respond specifically with regards to: Reviewer 2 comment ""High number of references are from online and popular media sources. Where is the data?"

This comment is interesting, however as issues with regards to decentralization and the BRI are so recent, there are hardly any peer-reviewed publications on this topic, at least not according to our research, which is thus mainly based on "grey literature": government publications, white papers and newspaper articles.

Thus the novelty and importance to research and publish on this topic, which is what we are intending with the revised commentary. We hope the rewritten brief communication meets expectations. Thank you once again to NHESS for considering our revised manuscript for publication.

---

## Referee Report (RR1)

NHESS review

This revision is a significant improvement on the previously submitted draft. However, in several places the central argument as well as the data analyzed can and should be presented more explicitly. My comments below have three main parts:

1. This central point made here should come at the start of the article so that the reader knows exactly what is being argued. The data analyzed can then be presented in more direct relation to substantiate this claim: "This commentary suggests that the issue of poor roads in Nepal is a political, not a technical issue and one where better service and less environmental damage could both be significantly addressed through improved governance."

2. While statements like this are partially true – "In general, connectivity is thus positively correlated with lower poverty rates" – other research shows that roads in fact increase levels of social stratification, marginalization, and uneven development. In addition to these references (Hettige, 2006; Iimi et al., 2016) which are cited numerous times at the start of the article, this reviewer strongly advises the authors to engage more extensively (one paragraph at least) with the highly influential work conducted by the University of East Anglia research team in the 1970s and 1980s (Blaikie et al. 1976). Please also double-check the citation year for this reference in the text. Please note that Rankin et al. 2017 closely review the findings of these studies as well.

3. The BRI issue is highly important but in the current version the text touches on the topic very lightly. I would suggest taking one of two alternative approaches: 1. discuss the BRI and its significance in more depth (not only what it means for Nepal, but why it has been taken up with such enthusiasm by elites in KTM, as well as the ambiguous and discursive nature of the BRI – a reified 'thing' that thus far has no real 'thingness'); or 2. pay less attention to the BRI and instead focus on the connection between road construction, landslides, and increasing risks and hazards due to climate change. I think the latter (#2) is actually a far more important intervention that this article can make to the current literature and broader knowledge of road construction and landscape change and hazards in the current political and climatic environments. BRI gets lots of attention these days, but this paper is not saying all that much new or contributing a great deal to such conversations. Conversely, by building on Petley, etc., it has much to offer to debates around the connections between road construction and landslide frequency.

I hope these comments are helpful with the next round of revisions and I look forward to reading the final version.

---

## Author Response (AR2)

Reviewers' comments v2

November 5, 2018

This revision is a significant improvement on the previously submitted draft. However, in several
places the central argument as well as the data analyzed can and should be presented more
explicitly. My comments below have three main parts:

Sudmeier : Authors thank the anonymous reviewer for these points. We have done our best to
incorporate them in the final version of the article
1. This central point made here should come at the start of the article so that the reader knows
exactly what is being argued. The data analyzed can then be presented in more direct relation to
substantiate this claim: "This commentary suggests that the issue of poor roads in Nepal is a
political, not a technical issue and one where better service and less environmental damage could
both be significantly addressed through improved governance."

Sudmeier:  Good suggestion. We have corrected accordingly.
2. While statements like this are partially true – "In general, connectivity is thus positively
correlated with lower poverty rates" – other research shows that roads in fact increase levels of
social stratification, marginalization, and uneven development. In addition to these references
(Hettige, 2006; Iimi et al., 2016) which are cited numerous times at the start of the article, this
reviewer strongly advises the authors to engage more extensively (one paragraph at least) with
the highly influential work conducted by the University of East Anglia research team in the 1970s
and 1980s (Blaikie et al. 1976). Please also double-check the citation year for this reference in the
text. Please note that Rankin et al. 2017 closely review the findings of these studies as well.

Sudmeier: The reviewer rightly points out the important work that was undertaken by this group in
the 1970s and 80s.  Although our focus is more on environmental impacts of roads, rather than on
socio-economic impacts, we have drafted some text to demonstrate that our ideas build on
previous work in this domain.
3. The BRI issue is highly important but in the current version the text touches on the topic very
lightly. I would suggest taking one of two alternative approaches: 1. discuss the BRI and its
significance in more depth (not only what it means for Nepal, but why it has been taken up with
such enthusiasm by elites in KTM, as well as the ambiguous and discursive nature of the BRI – a
reified 'thing' that thus far has no real 'thingness'); or 2. pay less attention to the BRI and instead
focus on the connection between road construction, landslides, and increasing risks and hazards
due to climate change. I think the latter (#2) is actually a far more important intervention that this
article can make to the current literature and broader knowledge of road construction and
landscape change and hazards in the current political and climatic environments. BRI gets lots of
attention these days, but this paper is not saying all that much new or contributing a great deal to
such conversations. Conversely, by building on Petley, etc., it has much to offer to debates around
the connections between road construction and landslide frequency.

Sudmeier: The previous draft was more heavily focused on the BRI and its potential influence.
However as rightly pointed out by the reviewer, for now the BRI is a 'thing' with no concrete plans
for Nepal yet, according to our understanding.  We will therefore go with option 2 and have
hopefully modified the manuscript accordingly.
We hope these comments are helpful with the next round of revisions and I look forward to reading
the final version.

Sudmeier:  Once again, we thank the anonymous reviewer for helpful and detailed comments
which have significantly improved the quality of this manuscript.  We also thank the guest editor
for useful suggestions and guidance throughout the process.

N.B. Figure 1 :  we need to remove the top line of the legend.  The colleague who made this map is
out of reach until after 15 November.  Thank you for your understanding.  Karen Sudmeier

[revised manuscript text omitted]

---

## Author Response (AR3)

Dear Bruce,

Thank you for accepting our manuscript – we are also very grateful that it has been given the status
"invited paper". I appreciate your additional suggestions which I have duly incorporated in this final
version. I just had one question inserted below with regards to figure 2.

Technical changes:

* Please check that all references cited are in the reference list and all references in the reference list
are cited at least once. I did a spot check and found items such as Iimi et al. (2016) line 83 was not in
the references. If you could check all, that would be helpful.

KSR: Iimi was in the references but I found other errors which I have corrected

* Can you do a last double check that all sentences with ideas/facts have appropriate in-text citations
or refers the reader to later in the manuscript where in-text citations are mentioned? For example,
line 97-98 discuss 'building on research and publications' but then does not give citations or direct
the reader to other parts of the manuscript.

KSR: I have reviewed and made a few corrections.

* Please ensure that three authors are first author surname and then et al. (line 100, this should be
Blaikie et al. not listing all three surnames).

KSR: Corrected

* Line 103, starting a new paragraph, it is a little confusing saying 'this work'. I suggest you state "The
work presented in this paper"

KSR: Corrected

* For SI units, you do not need to explain what they are the first time you put them in. Line 136
kilometers can be left as just km.

KSR: Corrected

* Figure 2—can you put a scale or give us an idea of scale in the figure caption. Is there a credit?

KSR: I have added a scale. The credit was already in the caption – do you prefer it is inserted in the
photo?

* In-text citations—when you have more than one in-text citation, you sometimes do them by date
(oldest to newest), other times alphabetically by surname. I suggest the former, but please be
consistent.

KSR: Corrected by chronological order

* In-text citations—line 172, avoid putting 'accepted' in the text, this should be (if necessary) in the
reference list and read 'in press' if not other information.

KSR: Corrected

I'd like to thank you very much for having chosen to publish with NHESS and again apologize for
delays. I look forward to seeing your manuscript in print.

KSR: Thank you once again for this opportunity to publish with NHESS, wishing you all the best

Karen

[revised manuscript text omitted]